# Sustainable production of highly conductive multilayer graphene ink for wireless connectivity and IoT applications

Kewen Pan[1], Yangyang Fan[2], Ting Leng[1], Jiashen Li[2], Zhiying Xin[2], Jiawei Zhang[1], Ling Hao[3], John Gallop[3], Kostya S. Novoselov[4,5] & Zhirun Hu[1,5]

Printed electronics offer a breakthrough in the penetration of information technology into everyday life. The possibility of printing electronic circuits will further promote the spread of the Internet of Things applications. Inks based on graphene have a chance to dominate this technology, as they potentially can be low cost and applied directly on materials like textile and paper. Here we report the environmentally sustainable route of production of graphene ink suitable for screen-printing technology. The use of non-toxic solvent Dihydrolevoglucosenone (Cyrene) significantly speeds up and reduces the cost of the liquid phase exfoliation of graphite. Printing with our ink results in very high conductivity ($7.13 \times 10^4 \, \text{S m}^{-1}$) devices, which allows us to produce wireless connectivity antenna operational from MHz to tens of GHz, which can be used for wireless data communication and energy harvesting, which brings us very close to the ubiquitous use of printed graphene technology for such applications.

[1] School of Electrical and Electronic Engineering, University of Manchester, Manchester M13 9PL, UK. [2] School of Materials, University of Manchester, Oxford Rd, Manchester M13 9PL, UK. [3] National Physical Laboratory, Hampton Road, Teddington TW11 0LW, UK. [4] School of Physics and Astronomy, University of Manchester, Manchester M13 9PL, UK. [5] National Institute of Graphene, Manchester M13 9PL, UK. Correspondence and requests for materials should be addressed to Z.H. (email: z.hu@manchester.ac.uk)

Development of printed conductive inks for electronic applications has grown rapidly due to widening applications in transistors[1], sensors[2], antennas[3,4], radio frequency identification (RFID)[5–7] tags, wearable electronics[8], etc. As conductor is the core component in printed electronics, efforts have been mainly focused on providing highly conductive metal nanoparticle inks, especially using silver nanoparticles[9]. Silver nanoparticle inks are widely used because of their high conductivity and good levelling property. However, silver is too expensive to be employed for low-cost applications[10]. Other metal nanoparticles such as copper or aluminium are much cheaper but can be easily oxidized. To avoid oxidation, a sintering process is necessary after printing[11,12]. However, in the case of heat-sensitive substrates (paper, plastic, etc.), sintering cannot be applied. Conductive polymers can also be fabricated as conductive film. This technique is however both chemically and thermally unstable[13]. Carbon nanotubes (CNTs) were once seen as an alternative for metal nanoparticles. The extremely high junction resistance between CNTs results in lower surface conductivity and hinders their applications[14]. The printed electronics industry has long been crying out for low-cost highly conductive inks.

Graphene ink, a dispersion of graphene flakes in solvents, can be easily patterned via spraying[15], screen printing[16], inkjet-printing[17,18] and doctor-blading[19] techniques. For antenna printing applications, spraying is a less reported method, suffering from lacking of flatness in films. Inkjet-printing and doctor-blading methods are complementary, the former having high accuracy and cost, in contrast to the latter. In addition, inkjet-printing has to print many cycles[20] to achieve low sheet resistance, which costs time and is not economically viable for mass production. Considering costs, printing accuracy and surface conductivity, the screen printing technique is the best candidate for industrial scale production. However, few screen-printed graphene devices have been reported including electrodes[13,21,22], electronic circuits[23] and antennas[16,19,24]. Most graphene ink processes use organic solvents such as N-methyl-2-pyrrolidone (NMP) and dimethylformamide (DMF). They are toxic, with low concentrations, and unsustainable, preventing them from using for industrial scale production.

Here we report the use of a cellulose derived solvent, dihydrolevoglucosenone (Cyrene), which is not only non-toxic, environmentally friendly and sustainable but also can provide higher concentration of graphene ink, resulting in significant cost reduction for large-scale production. In this work, low cost, environmentally friendly and sustainable, highly conductive graphene ink (10 mg mL$^{-1}$) has been developed and was further concentrated to 70 mg mL$^{-1}$ for screen printing. More importantly, we have demonstrated that printed graphene antennas, ranging from high-frequency band (a few tens of MHz) to microwave band (a few tens of GHz), can be applied across the entire RF spectrum. As critical demonstrations, a printed graphene-enabled battery-free wireless body temperature sensor, RFID tags and RF energy harvesting system for powering battery-free devices that are capable of sensing resistive and capacitive sensors are presented, illustrating the potential of low cost, screen-printed graphene enabled wearables for IoT applications such as healthcare and wellbeing monitoring, also embodying sustainability and disposability, all of these are critical factors to enter the mass-produced market.

## Results

### Ink characterization.
Conductive graphene ink has been researched for a number of years. It is now possible to obtain defect free, less oxidized and stable graphene flakes by liquid phase exfoliation[25,26] which can be deposited on different substrates. Many organic solvents with specified surface energy[27] have been verified for graphene exfoliation under bath sonication treatment with low residual and better stability such as NMP and DMF[27,28]. However, low concentrations, environmentally harmful and toxic properties of these organic solvents have prevented their applications from industrial scale graphene ink production[29]. Alternative method is to exfoliate graphene in low cost aqueous-based solutions with surfactants[30–32]. A recent work[33] proposed ultrahigh concentration (50 mg mL$^{-1}$) graphene slurry in water, but oxidization on the edge of graphene flakes still degrades its conductivity.

Bio-based Cyrene (CAS: 53716-82-8) was first identified as a high-performance solvent for graphene dispersion in 2017[29]. The solvent not only has appropriate polarity[29,34] and surface tension[27,29,35] which are especially suitable for graphene ink preparation, but also the concentration of dispersion where graphene can exist stably is at least an order higher than other organic solvents[29]. Moreover, Cyrene is non-toxic and easily extracted from cellulose which is abundant on Earth, has a great potential for low cost, environmentally friendly and sustainable industrial scale graphene ink production. However, for printed electronic applications, the most crucial property, electrical conductivity of Cyrene-based graphene ink, has not been reported, nor have its electronic applications.

In this work, expanded graphite was added into Cyrene and NMP (as comparison). Graphene flakes were produced during sonication treatment. Firstly, the sonication time for exfoliation was investigated as it is of significance for large-scale ink production. Samples were extracted at different sonication time. It is easy to remove unexfoliated graphite particles in extracted samples via centrifugation and filtration (see Methods). It has been noticed that long sonication period also affects the quality of graphene flakes and degrades the conductivity of graphene[36]. For wireless connectivity applications, the conductivity of the printed graphene pattern is crucially significant. Thin graphene flakes allow for the best stacking, however, they end up with the maximum number of interfaces, which potentially can increase the resistance. Thick graphene flakes allow one to reduce the number of interfaces between the flakes, but they do not guarantee good stacking and result in a number of voids when printed. The conductivity can be maximized by selecting flake thickness, otherwise low conductivity increases connection loss and jeopardizes the ink applications. To evaluate conductivity, the sheet resistance variations of the graphene laminate under different sonication time in both NMP and Cyrene solvents were plotted in Fig. 1a. As it can be seen, the sheet resistance of the graphene laminate decreases rapidly within the first 8 h ultrasonic exfoliation in Cyrene whereas it takes 20 h in NMP to reach the similar sheet resistance. The rate of decreasing of sheet resistance with Cyrene ink is significantly faster than that of NMP ink (red and black dash lines), which exceeds 0.6 Ω sq$^{-1}$ h$^{-1}$ between 4 and 8 h. The minimum sheet resistance of Cyrene ink is 0.78 Ω sq$^{-1}$, which happens at 8 h sonication. After that, the sheet resistance rises as sonication time further increases because the longer sonication time leads to smaller flake sizes as well as damages sp$^2$ network of graphene flakes[36], affecting the graphene flake's electrical conductivity, which is not recoverable. For NMP solvent, the decreasing rate of sheet resistance is slower. The minimum sheet resistance is 0.77 Ω sq$^{-1}$, happens at 48 h sonication. That said, although the physical properties of both solvents are quite similar[29], the best sonication time for NMP is six times longer than that for Cyrene, which indicates that Cyrene has better exfoliation efficiency compared to NMP, providing an advantage in time saving and cost reduction for large-scale graphene ink production.

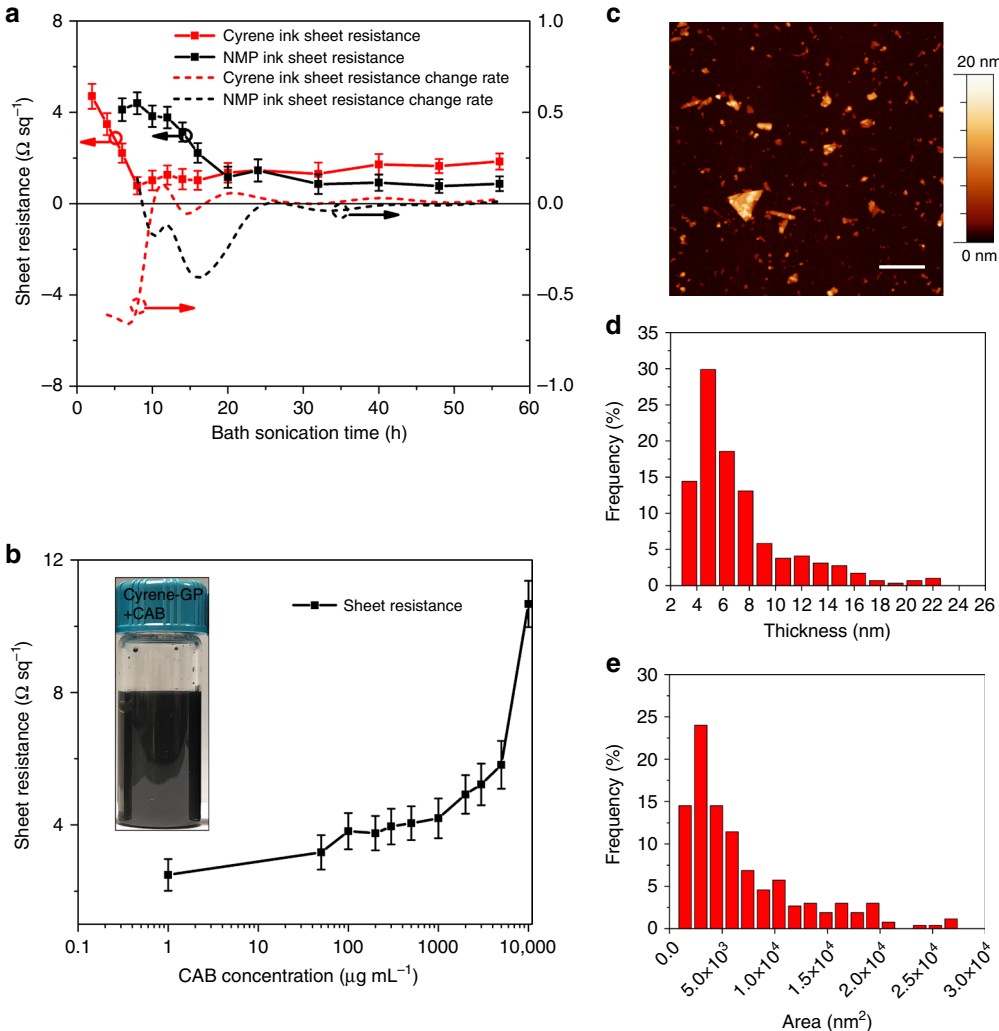

**Fig. 1** Quality of exfoliated graphene flakes in Cyrene. **a** Measured average sheet resistance values (left axis, measured five times per point) and variation (right axis) of sheet resistance as a function of sonication time (black line: NMP, red line: Cyrene). **b** Sheet resistance variation (measured five times per point) with different CAB concentrations and the insert sample of 10 mg mL$^{-1}$ graphene ink with 1 mg mL$^{-1}$ CAB. **c** AFM image of graphene flakes on silicon substrate; scale bar is 1 μm, **d** thickness histograms and **e** flake size

The electrical conductivity of dried and compressed graphene laminate (8 h exfoliated in Cyrene, see in Methods) is $7.13 \times 10^4$ S m$^{-1}$, higher than any work reported so far[37–39], confirming that Cyrene is an excellent solvent with higher exfoliation efficiency and less defects on graphene flakes for replacing traditional toxic organic solvents.

Binder-free graphene ink and its applications have been reported[16,40]. However, the adhesion was less impressive. For practical applications, adhesive materials are normally added in the ink. The drawback for adding adhesive materials is that it will significantly degrade the ink electrical conductivity. In this work, cellulose acetate butyrate (CAB) was added into the Cyrene graphene ink, acting as a polymer-assisted agent. CAB can stabilize the ink because the electrostatic repulsion between CAB molecules prevent graphene flakes from restacking and aggregation[41], also enhance adhesion and anti-scratching performance of the printed pattern. However, addition of CAB to the ink decreases its conductivity dramatically. In order to understand the relationship between CAB concentration and sheet resistance, 0.05, 0.1, 0.2, 0.3, 0.4, 0.5, 1, 2, 3, 5 and 10 mg CAB was dissolved into 1 mL of pristine graphene ink (10 mg mL$^{-1}$, exfoliated in Cyrene for 8 h) with a short period of sonication treatment. As can be observed in Fig. 1b, the sheet resistance of the graphene/

CAB laminate rises relatively rapidly when CAB concentration is less than 100 μg mL$^{-1}$ but slowly increases while the CAB concentration is between 100 μg mL$^{-1}$ and 1 mg mL$^{-1}$. The sheet resistance is about two times higher than that of pristine graphene laminate when the concentration of CAB is 1 mg mL$^{-1}$. The sheet resistance increases logarithmically for CAB concentration over 1 mg mL$^{-1}$. To achieve good conductivity as well as printing quality, 1 mg mL$^{-1}$ CAB concentration was applied in this work. The inset in Fig. 1b shows the ink sample.

Atomic force microscopy (AFM) was used to characterize graphene flakes (prepared from graphene/CAB ink, 10 mg mL$^{-1}$ with 8 h sonication). Clear graphene flakes are shown in Fig. 1c with a lateral area of $6 \times 6$ μm, which confirms the stable existence of few-layers graphene nanoflakes in the high concentration ink (AFM image of a single graphene nanoflake profile can be seen in Supplementary Figure 1). The measured flake thickness and size distribution (291 flakes were counted) are peaked at 5 nm (Fig. 1d) and $2.5 \times 10^3$ nm$^2$ (Fig. 1e), respectively. It is worth noticing that the statistics follow lognormal distribution of high power sonication of 2D materials as expected[42].

Fourier transform infrared spectroscopy (FTIR) and Raman spectroscopy were used to investigate the quality of exfoliated graphene flakes. Fig. 2a illustrates the FTIR spectra of pure

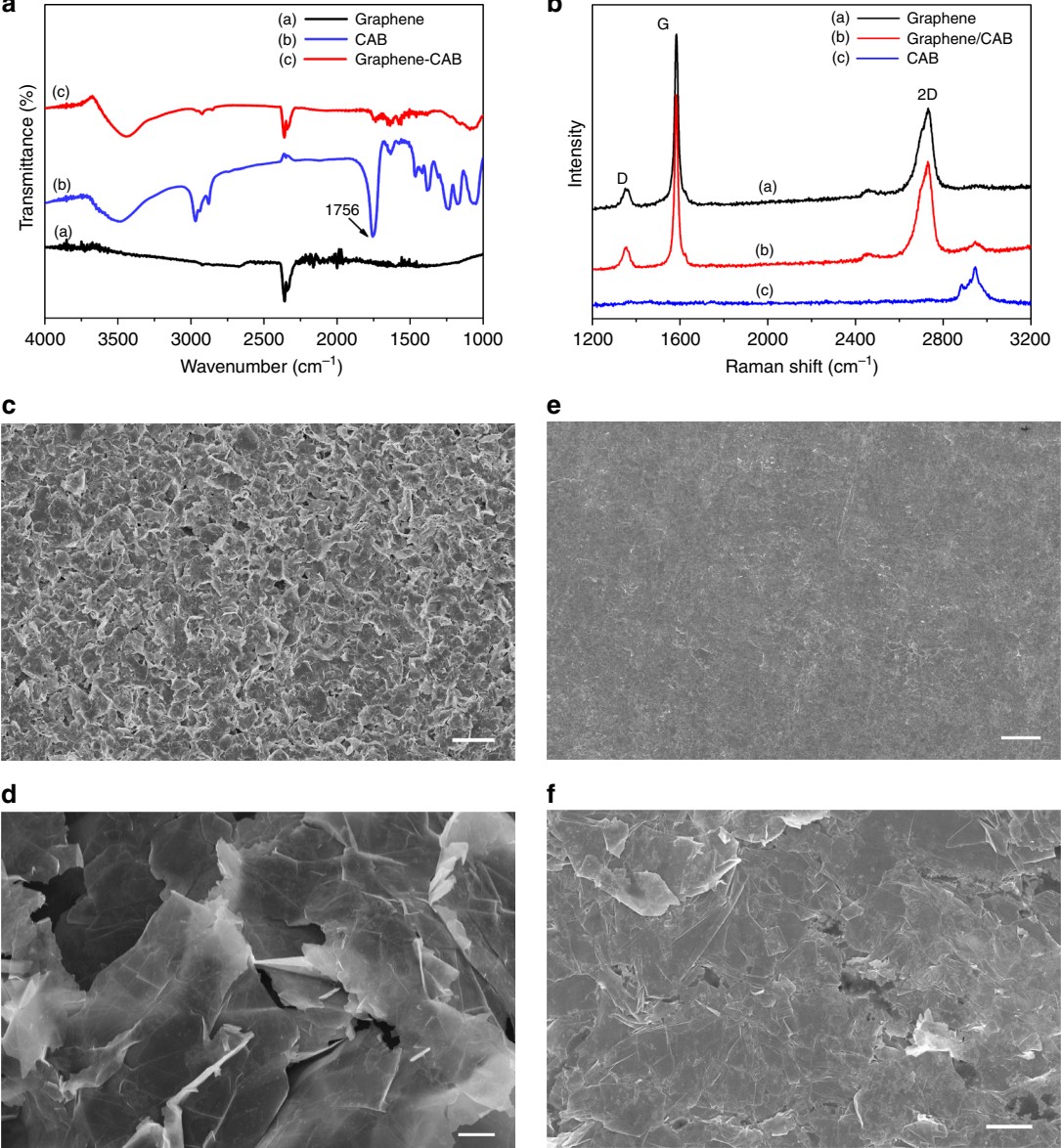

**Fig. 2** Quality of the screen-printed graphene laminate: **a** FTIR characterization of Cyrene graphene ink with and without CAB, **b** Raman spectra of Cyrene graphene ink with and without CAB and **c–f** SEM images of screen-printed graphene on paper (**c** uncompressed and **e** compressed screen-printed graphene laminates with ×300 magnification; scale bar is 30 μm, **d** uncompressed and **f** compressed screen-printed graphene laminates with ×10k magnification; scale bar is 1 μm)

exfoliated graphene, exfoliated graphene with CAB assisted and CAB itself. Compared to reduced graphene oxide (rGO) or chemically derived graphene, it is worth noticing that no peaks associated with −OH (∼1340 cm$^{-1}$) and −COOH (∼1710−1720 cm$^{-1}$) groups are detected for the exfoliated graphene[43] (black line in Fig. 2a). The absence of peaks is evidence that the graphene flakes is composed of largely defect-free material. CAB is a polymer that has been partially esterified but still has large numbers of hydroxyl groups which have broad OH absorption located at 3490 cm$^{-1}$ (blue line). A characteristic peak is seen at 1756 cm$^{-1}$ attributed to CAB (C=O stretch) which can also be observed on graphene/CAB (red line). Raman spectroscopy of exfoliated graphene is shown in Fig. 2b, featuring the breathing mode of sp$^2$ carbon atoms at D-band (1355 cm$^{-1}$), G-band (1583 cm$^{-1}$), associated with in-phase vibrations of the graphite lattice, the relatively wide 2D-band at 2731 cm$^{-1}$ and an overtone of the D-band[27,44]. A low D/G ratio indicates fewer defects on graphene

flakes[45], which is significant for electron flow and no structure change of graphene flakes can be detected by Raman in the graphene/CAB sample.

Morphological features of graphene/CAB laminate were investigated by using a scanning electron microscope. Graphene flakes can be clearly seen at Fig. 2c, d. Uncompressed graphene is curly and has poor adhesion between flakes. It is obvious that graphene flakes were randomly stacked to each other. There are gaps (dark holes) between the flakes, severely degrading the contact quality. Around the gaps, electron flow between the graphene flakes appears between the edges and tips of the flakes, which results in a relatively large sheet resistance (37 Ω sq$^{-1}$). Hence, the following compression process is significant for improving sheet resistance. A paper rolling machine (Agile F130 Manual Mill) was used to compress the printed patterns. As it can be seen in Fig. 2e, f, the surface is no longer coarse after compressing, graphene flakes are piled sequentially with face-to-

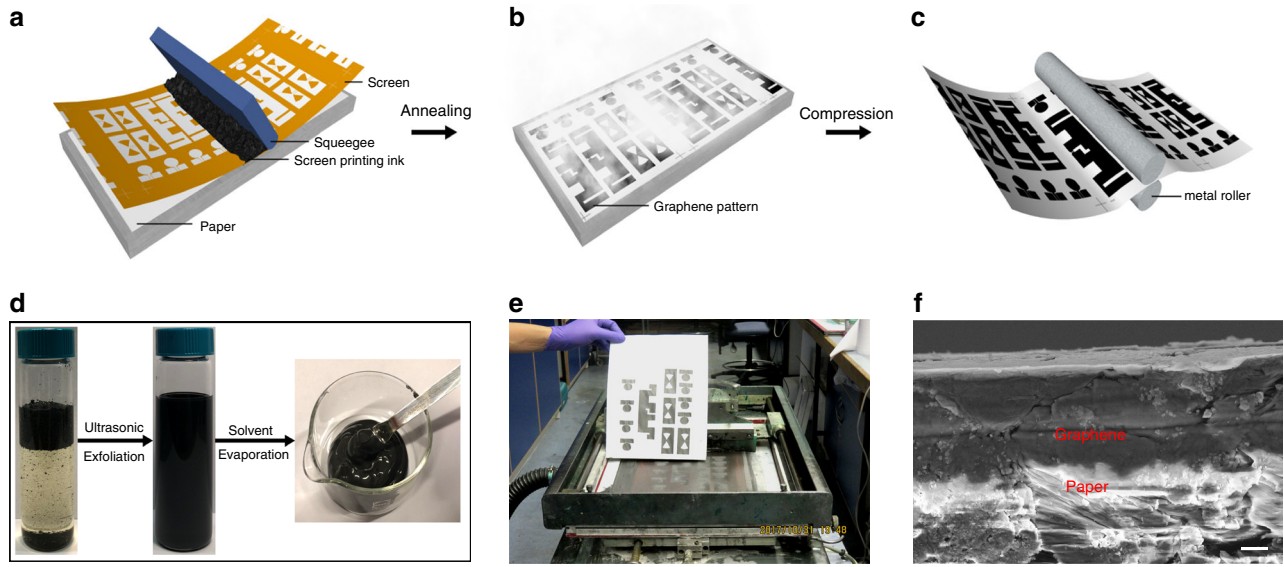

**Fig. 3** Graphene antenna fabrication using screen printing technology. Screen-printing steps: **a** Patterning graphene ink via exposed screen and squeegee, **b** annealing printed patterns and **c** compressing dried pattern with steel rolling machine. **d** Cyrene-based graphene ink and high concentrated (70 mg mL$^{-1}$) screen printing ink. **e** Demonstration of printed antennas on A4 paper. **f** SEM cross-sectional view of the printed graphene antenna; scale bar is 1 μm

face contacts, dramatically reducing the sheet resistance. The sheet resistance of compressed patterns was measured to be 1.2 Ω sq$^{-1}$ on average, 30 times smaller than uncompressed patterns.

**Antennas design and fabrication.** A commercial manual screen printer was used in this work. Fig. 3a–c demonstrates the straightforward steps of graphene antenna screen-printing: Fig. 3a graphene ink is uniformly added on the exposed screen with negative antenna patterns and a squeegee is moved from one side to another, transferring the ink on to the substrate, Fig. 3b thermal annealing and Fig. 3c compression. Ultrahigh graphene concentration (70 mg mL$^{-1}$) screen-printing ink was achieved via rotary evaporation from 10 mg mL$^{-1}$ graphene/CAB ink, as seen in Fig. 3d (the viscosity data can be found in Supplementary Figure 2). The screen printer and printed graphene antenna patterns are illustrated in Fig. 3e. The resolution of our screen-printed patterns is 0.4 mm. A cross-sectional view of the screen-printed graphene/CAB laminate on paper is shown in Fig. 3f. There is no obvious boundary between graphene and paper, revealing good adhesion. The printed patterns have excellent mechanical flexibility, as shown in Fig. 4a. Such flexible property has a great potential in wearable, deformable IoT applications[46]. The average thickness of screen-printed graphene laminate is 7.8 μm and conductivity can therefore be calculated as $3.7 \times 10^4$ S m$^{-1}$, which is approximately about half of that of the pristine graphene ink (without CAB) but still has about the same conductivity as recently reported work[47] which however requires high temperature (350 °C) annealing. Additional bending and adhesion performance of the printed graphene/CAB pattern were tested. Low residual blue tape (BT-150E-KL) was used to test adhesion performance by sticking and peeling off from printed pattern repeatedly[47]. The sheet resistance of pristine graphene pattern doubled after first cycle. In the meantime, the sheet resistance of graphene/CAB pattern increased 5%. After 10 cycles, the sheet resistance of graphene/CAB pattern increased 40% whereas pristine graphene pattern was peeled off from paper completely and its sheet resistance became too large to be measured (>20 MΩ sq$^{-1}$) at cycle 7. In a bending test, a 3 cm × 1 cm rectangular pattern was printed and compressed. The pattern was bent from 0° to 90° for 2000 cycles while its resistance only increased 5%, which is comparable to other published works[48].

Three different types of antennas were printed, ranging from near field communication (NFC; Fig. 4b), ultrahigh frequency (UHF; Fig. 4c), RFID to C-X-K$_u$ ultra-wideband slot antennas (Fig. 4d). These antennas were designed for low cost, flexible and disposable wireless applications. For instance, the environmentally friendly printed graphene NFC antenna can replace traditional metal NFC antenna for access card applications. The wideband slot antenna can replace metal antennas for ultra-wideband data communication with conformability and lower cost. All antennas were designed and simulated by using commercial full-wave electromagnetic simulation software CST[49]. In the simulation, the printed graphene/CAB laminate was modelled as ohmic sheets because the laminate thickness is much smaller than its skin depth. One of the beauties of using graphene ink to print antennas is that the sheet resistance can be controlled from one to tens of ohms per square. This provides extra design freedom depending on the antenna applications. For high gain, high efficiency antennas, lower sheet resistance will be required. For wideband antennas, however, relatively higher sheet resistance ink can be used if the radiation efficiency is not critical. The geometrical dimensions of the antennas are all illustrated in Fig. 4b–d (sheet resistance details can be found in Supplementary Figure 3). The inner line and gap width of coplanar waveguide transmission line applied on wideband slot antenna is 3 and 0.4 mm respectively.

**NFC battery-free temperature sensor.** NFC technology plays an increasingly important role with the development of IoT technology. It not only can be applied for access or ID cards but can also be used for other near field wireless monitoring applications, such as wireless healthcare and wellbeing monitoring. In this work, a wireless body temperature monitoring system has been designed and demonstrated. The sensor tag consists of disposable graphene printed planar coil antenna, temperature sensor (NTHS0603N17N2003JE, VISHAY) and functional NFC chip (RF430FRL152H; Texas Instruments), as shown in Fig. 4e (more information can be found in Supplementary Figure 4 and Supplementary Note 1). The printed graphene NFC antenna harvests RF power for the chips and provides data communication when activated by the reader. The real-time temperature monitoring is illustrated in Fig. 4f at the distance of 2.5 cm between tag and reader. The near field reader

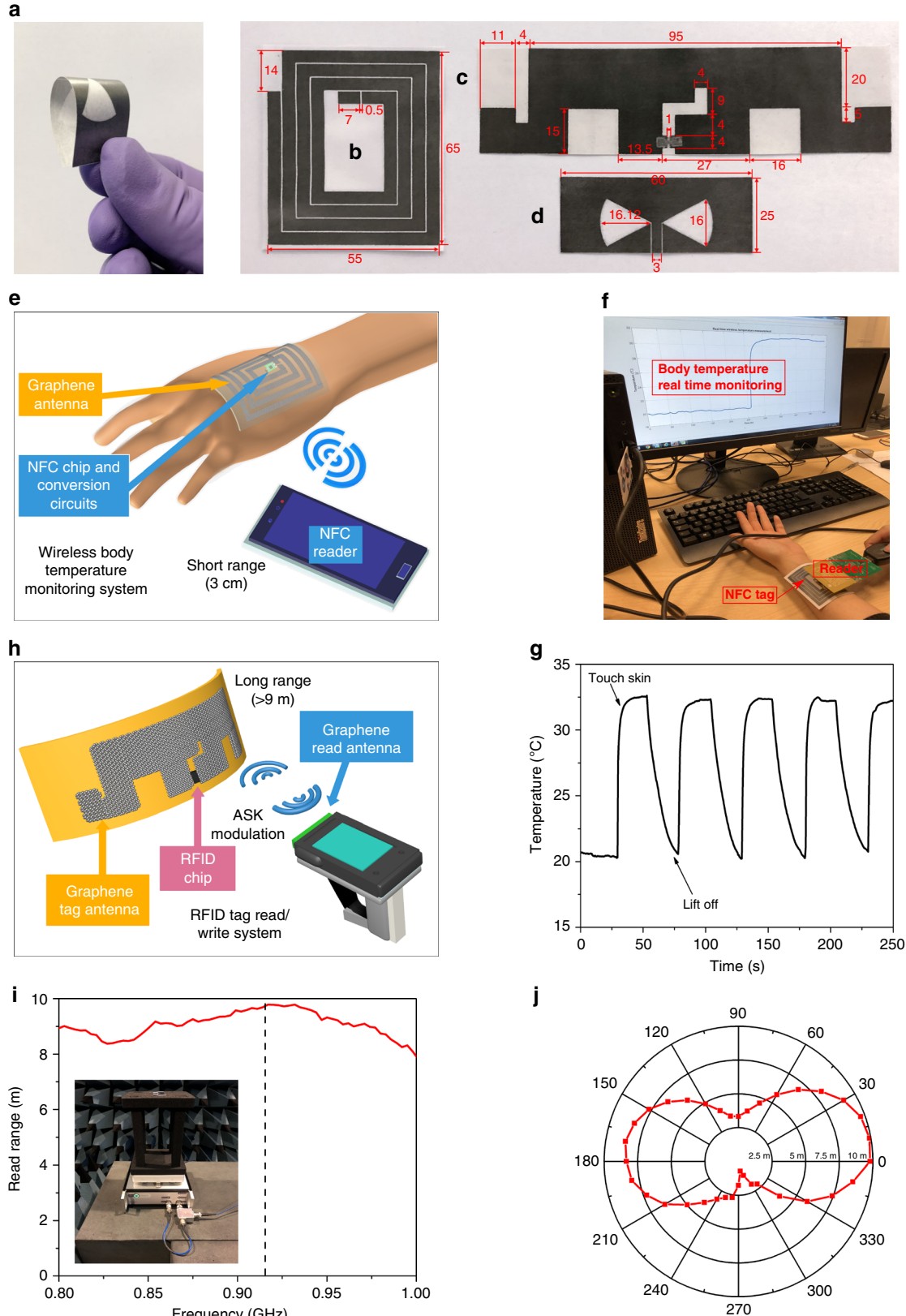

**Fig. 4** Printed graphene antennas and IoT applications. **a** Flexibility of the printed graphene antenna. **b–d** Printed graphene antennas' geometric parameters (mm): **b** NFC antenna (without NFC chip and jumper), **c** UHF RFID antenna, **d** wideband slot antenna. **e–g** Healthcare applications (**e** Illustration of graphene printed NFC temperature sensing system, **f** Demonstration of measurement and **g** recorded data of body temperature). **h–j** UHF RFID tag applications demonstration (**h** Illustration of printed graphene RFID antenna system, **i** Read range and **j** radiation pattern (E-field, at 915 MHz))

(TRF7970AEVM; Texas Instruments) communicates with the sensor tag for continuously recording body temperature and uploads data to a designated terminal. Fig. 4g illustrates a data frame that was transmitted from the near field temperature sensor. There is no transmission error in 250 s measurement period. A clear, repeatable high- and low-temperature variation can be observed when the near field sensor is placed on the human skin and removed from it. This can be very useful for wireless monitoring patients' temperature in hospital wards and even at home. The data can then be relayed to the cloud and analysed by professional health workers remotely.

**Long read range UHF RFID antenna.** In order to further demonstrate the potential of printed graphene antennas, a UHF RFID antenna has been designed, optimized and printed for long read range communication, as shown in Fig. 4h. The antenna tag consists of a radiator (printed graphene), T-matching network (printed graphene) and an RFID chip (Impinj Monza R6). One of the most important technical merits for a UHF RFID antenna tag is its read range $r$, which can be calculated by[50]

$$r = \frac{\lambda}{4\pi} \sqrt{\frac{P_t G_t G_r \tau}{P_{th}}}, \quad (1)$$

where $\lambda$ is the wavelength, $P_t$ is the transmitted power from the reader antenna, $G_r$ is the gain of the reader antenna, $G_t$ is the gain of tag antenna, $P_{th}$ is the minimum threshold of the power needed to activate the RFID chip and $\tau$ is the matching factor, which varies from 0 to 1 and is given by[51]

$$\tau = \frac{4R_c R_a}{|Z_c + Z_a|^2} \quad (2)$$

where $Z_c$ and $Z_a$ represent chip impedance and antenna input impedance, respectively. $R_c$ and $R_a$ are the real parts of the chip and antenna impedance. Conjugate impedance matching is required between the RFID antenna and the chip in order to maximize the matching factor, resulting in maximal read range. The maximum read range of the printed graphene UHF RFID antenna vs. frequency is illustrated in Fig. 4i, which shows that a 9.8 m read range at 917 MHz was achieved and the tag has long read range of over 9 m from 854 to 971 MHz, covering the whole UHF RFID band (860–960 MHz). This is a very useful property as it provides enough frequency shift redundancy for printing tolerance and different application environments. This result, doubling the read range of the nearest work in printed graphene RFID[19], rival to aluminium etched commercial RFID antennas and silver ink printed RFID ones[52]. For a more intuitive demonstration, the radiation pattern in the E-plane at 915 MHz was plotted against maximum read range instead of antenna gain and the data was recorded for every 10° rotation, as shown in Fig. 4j. A typical dipole pattern can be seen from the radiation pattern where the maximum reading range occurs at 0° and 360°, the minimum read range happens at 90° and 270°.

**Ultra-wideband antenna and energy harvesting applications.** Fig. 5a shows the reflection coefficient ($S_{11}$) of the printed graphene ultra-wideband slot antenna. The 10 dB bandwidth is from 3.8 to 15.5 GHz, achieving more than 120% fractional bandwidth. This wideband characteristic is very useful for upcoming 5G mobile communication and ultra-wideband radar applications[53]. The fundamental resonance of the slot antenna is around 5 GHz and low reflection extends close to 9 GHz. Above 9 GHz, the higher resonance modes start to play the major role. The fundamental and higher mode resonances overlapped around 9 GHz,

resulting in a wide bandwidth (simulated surface current distributions on the antenna at 4 and 12 GHz can be seen in Supplementary Figure 5a). Antenna gain was measured at the maximum gain point and shown in Fig. 5b. The antenna gain varies from 2.5 to 6 dB from 4.6 to 13.5 GHz. The radiation patterns for the printed graphene ultra-wideband slot antenna under different frequencies are illustrated in Fig. 5c–f. All data were plotted with linear form for comparison. As the first two radiation patterns at 4 GHz (Fig. 5c) and 8 GHz (Fig. 5d) show a typical symmetrical dipole pattern, this demonstrates that the antenna is working in its fundamental resonance mode. The gain maximum occurs at 0°. The radiation level at the front side is slightly stronger than the back side, probably due to the loss of paper substrate. For the radiation pattern at 12 GHz (Fig. 5e) and 14 GHz (Fig. 5f), typical dipole radiation patterns have disappeared, indicating higher mode resonance.

There are various wireless signals in free space at any moment of time. Such wireless energy can be harvested and stored to power up low-power, battery-free electronic devices intermittently. This strategy is becoming highly desirable due to the fast development of wireless sensor networks (WSNs) and IoT technologies[54,55]. Here we demonstrate a printed graphene-enabled RF energy harvesting system. The harvesting system consists of an ultra-wideband printed graphene slot antenna, low-pass filter that can suppress harmonic radiation[55] and rectifier with four-stage Cockcroft–Walton voltage multiplier which converts the RF to DC power and provides the required output voltage to drive low-power CMOS devices, as shown in Fig. 5g, h (the reflection coefficient of the conversion circuit can be found in Supplementary Figure 5b). To measure the conversion efficiency of the harvesting system, an RF signal generator and standard gain horn antenna which provides accurate RF energy and gain were used. The printed graphene antenna and conversion circuit were placed at 2 m away in an anechoic chamber. 5.8 GHz was applied in the measurement because more and more wireless devices (e.g. 5G WiFi, UAVs) are using this free license channel. The antenna total efficiency, circuit conversion efficiency and overall RF-to-DC conversion efficiency of the system were measured and plotted in Fig. 5i. It can be observed that the printed graphene antenna has constant efficiency (51%) for different RF power levels since the antenna is a linear device. The overall RF-to-DC conversion efficiency varies with RF power due to the diodes' nonlinearity and reaches its maximum of 22% ($R_{load}$ = 10 kΩ), which can be further improved by using diodes with lower barrier height in the voltage multiplier. From the view of IoT applications, high quality printed 2D material sensors[56,57] can be embedded into the system with the same printing process. A prototype of RF powered, battery-free low-power square wave oscillator which converts the sensors' analogue outputs to frequency modulated signals is developed and demonstrated here. The oscillator consists of two CMOS NOR gates (SN74AUP1G02), variable resistor (resistive sensor) and variable capacitor (capacitive sensor), as shown in Fig. 5g, which can be powered wirelessly as long as the incident RF power is over −12 dBm. One of the generated square waveforms by the system with a frequency of 5 kHz and peak to peak voltage of 2 V can be observed from the oscilloscope in Fig. 5h, experimentally verifying that the printed graphene enabled RF energy harvesting system is fully functional.

## Discussion

Environmentally friendly, sustainable, low cost, highly conductive and concentrated screen printing graphene/CAB ink has been developed. High-quality pristine graphene sheets were exfoliated and dispersed in Cyrene with the concentration of 10 mg mL$^{-1}$.

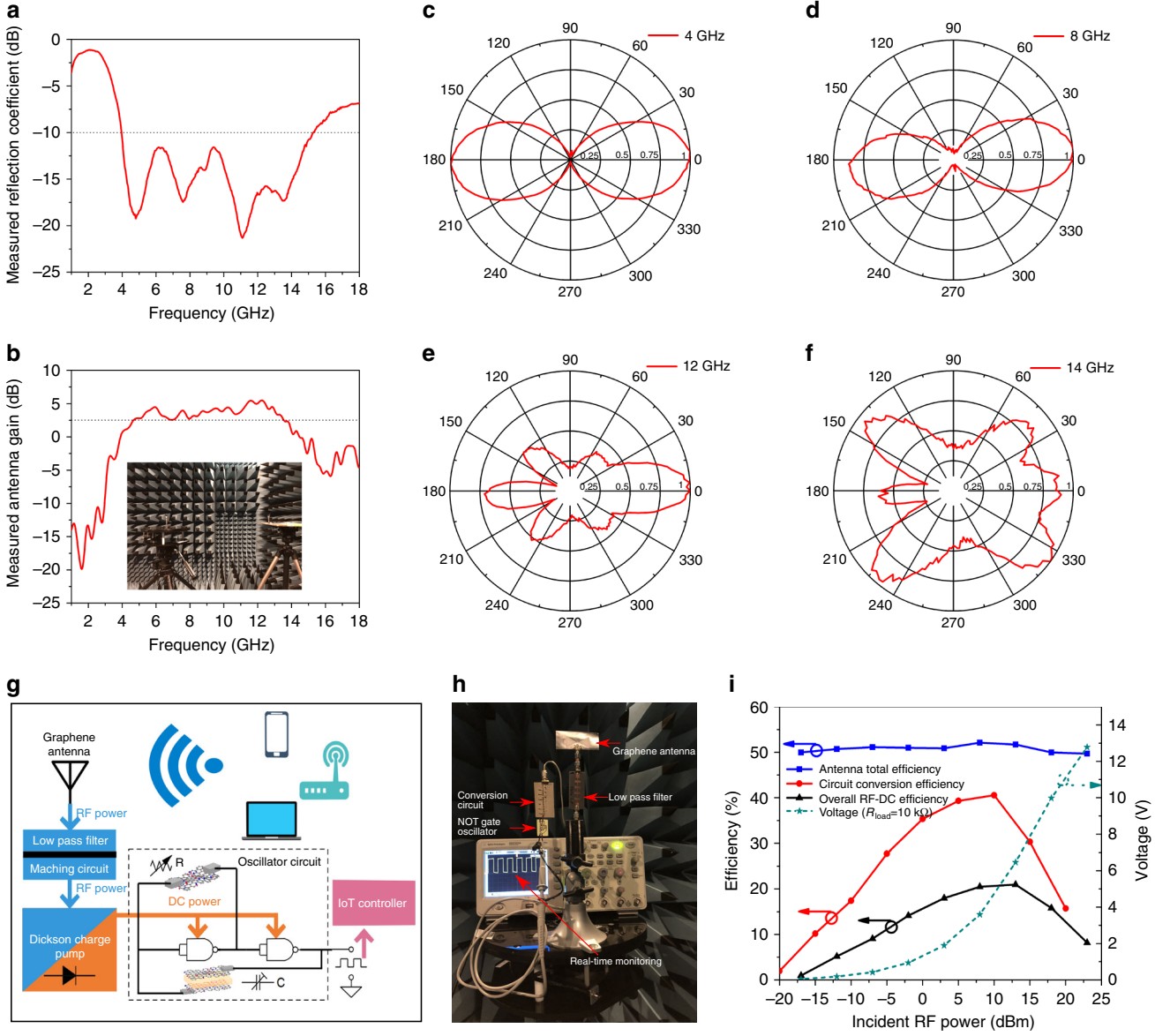

**Fig. 5** Printed graphene wideband slot antenna for RF energy harvesting application. **a** Measured reflection coefficient ($S_{11}$) of the slot antenna. **b** Measured antenna gain (three-antenna method). **c–f** Measured radiation pattern at 4 GHz (**c**), 8 GHz (**d**), 12 GHz (**e**) and 14 GHz (**f**). **g–i** Demonstration of RF energy harvesting application (**g** illustration of RF energy harvesting system, **h** measurement set up and **i** measured efficiencies and output DC voltage as a function of different RF power levels)

Using Cyrene has eliminated the use of toxic solvents, significantly simplifying post production treatments, especially propitious for industrial scale manufacturing. Exfoliation times and conductivity of Cyrene graphene ink were measured and compared to those of NMP one. Conductivity of $7.13 \times 10^4$ S m$^{-1}$ has been achieved after compressing, which is the highest reported so far. After adding CAB as a stabilizing agent and rotary evaporation, the further concentrated graphene ink (70 mg mL$^{-1}$) becomes screen printable. The printed graphene/CAB laminate still achieved a high conductivity of $3.7 \times 10^4$ S m$^{-1}$. The prototype NFC antenna in the high-frequency region, high-performance RFID antenna working in ultrahigh-frequency band and ultra-wideband antenna operating in microwave bands all provide evidence that printed graphene antennas can replace traditional metal antennas for wireless identification, sensing and data communications for low cost and ubiquitous wireless connectivity. Furthermore, graphene-based NFC temperature sensors for healthcare and wellbeing monitoring and graphene enabled energy harvesting system that can power battery-free CMOS oscillator have been successfully demonstrated, opening an avenue for low cost, environmentally friendly and sustainable printable devices at upcoming IoT applications.

## Methods

**Liquid exfoliation of graphene with high concentration**. Expandable graphite with +50 mesh flake size was purchased from Sigma-Aldrich. Cyrene (dihydrolevoglucosenone, >99%) was provided by Circa Group Pty Ltd. N-methyl-2-pyrrolidinone (NMP, >99%) was from Alfa Asia. CAB (butyryl content 35–39%) was from Arcos Organics. Graphene nanoflakes were obtained via the liquid phase exfoliation method. Expandable graphite were placed in a ceramic crucible and then heated in an 800 W commercial microwave oven for 30 s to obtain expanded graphite with fewer layers. The expanded graphite flakes were stirred and washed in deionized water to remove residual acid until the pH is close to 7. Then, the

mixture was dried in an oven for 5 h at 100 °C. Dried expanded graphite was deposited into NMP solvent ($10\,mg\,mL^{-1}$) and Cyrene ($10\,mg\,mL^{-1}$), respectively where these organic solvents provide appropriate surface energy for sonication processes. After that, the mixture was sonicated in an ultrasonic bath (SHESTO, UT8061-EUK). Samples were extracted at 0, 2, 4, 6, 8, 10, 12, 14, 16, 20, 24, 32, 40, 48, and 56 sonication hours for sheet resistance measurement.

**Screen-printing high concentration graphene ink preparation**. The exfoliated graphene nanoflakes were obtained in the dispersion after sonication. A 300-mesh stainless steel screen filtered the mixture first. After that, unexpanded graphite particles were removed after 5 min low-speed centrifugation (500 rpm). After that, the graphene dispersion was concentrated to $10\,mg\,mL^{-1}$ with vacuum rotary evaporation (Buchi R-114 Rotavap evaporator). CAB was added into Cyrene as a polymer-stabilizing agent to increase the stable existence of graphene flakes in Cyrene and improve the printing performance. In order to achieve appropriate viscosity for screen-printing, the graphene concentration of $70\,mg\,mL^{-1}$ was achieved (mixture of graphene flakes, CAB and Cyrene) by vacuum rotary evaporation. Before printing, the mixture was mechanically agitated for 5 min and bubbles inside the ink were removed after a short period of vacuum treatment. The printing screen with 62 mesh was specially fabricated with capillary film (ULANO, EZ50-Orange) to achieve uniform laminate of graphene paste. Then, the sample was dried and annealed in an oven (vacuum) at 100 °C for 5 h (comparison of antenna gains printed with Cyrene-based graphene ink (8 h sonication time) and NMP-based graphene ink (20 h sonication time) can be found in Supplementary Figure 6).

**FTIR**. Both of Graphene, CAB and Graphene-CAB samples were annealed in an oven (vacuum) at 100 °C for 48 h to ensure dryness and avoid errors of measurement. They were mixed with potassium bromide (KBr) with 1:20 mass ratio and ground in an agate mortar, respectively. FTIR data were recorded by a Fourier transform infrared spectrometer (Nicolet 5700).

**Raman**. Films of exfoliated graphene (without CAB), graphene/CAB ink and CAB were coated on $SiO_2/Si$ substrate. Raman spectra were acquired with a Renishaw System 1000 Raman Spectrometer. Micro-Raman spectrometer at a ×50 objective, with an incident powder of 2.3 mW. The system has within $1.5\,cm^{-1}$ spectral resolution at 514 nm.

**Atomic force microscopy**. Atomic force microscopy imaging of few layer graphene was measured with a Bruker Dimension Icon in tapping mode. The concentration of graphene/CAB ink is $10\,mg\,mL^{-1}$ and 100 μL ink sample was spin-coated (3000 rpm, 1 min) on a clean $SiO_2/Si$ substrate and washed by acetone, distilled water and isopropyl alcohol sequentially. The AFM sample was dried in an oven for 5 h at 100 °C.

**Scanning electron microscopy**. Scanning electron microscopy was performed using a ZEISS Sigma HV. Aperture size is 7 μm. The field emission gun is 5 KV and signal was acquired from InLens. For the cross-section measurement, the paper with printed graphene on top was soaked in liquid nitrogen and cut by a scalpel.

**Conductivity measurement**. Sheet resistance is used to describe the thin film resistance regardless of film thickness, it can be written as

$$R_s = \frac{\rho}{t}, \qquad (3)$$

where $R_s$ is sheet resistance, $\rho$ is the resistivity of the film and $t$ represents the thickness of film. For the measurement of sheet resistance variation with different ultrasonic treatment times, the ink was first filtered via a 300-mesh stainless steel screen. Small unexpanded graphite particles were removed after 5 min low-speed centrifugation (500 rpm, 5 min). To avoid coffee-ring effect, a filter paper (Whatman qualitative filter paper, Grade 5) was held on a glass funnel (140 mL Aldrich Buchner funnel) by vacuum. After that, 60 μL of the filtered sample was dropped on filter paper. The ink was annealed in an oven for 5 h (100 °C) and compressed with the rolling machine. The sheet resistance was measured using 4-point probe station (Jandel, RM3000) and semiconductor characterization system (Keithley, 4200C).

**Antenna reflection coefficient measurement**. For reflection coefficient measurement ($S_{11}$), the ultra-wideband slot antenna was directly connected to a calibrated Vector Network Analyzer (VNA, Fieldfox N9918A, Keysight).

**Radiation pattern measurement**. There are three graphene printed antennas to measure gain by themselves. The SMA (RS PRO, 526–5763) connector was connected with the antenna using conductive epoxy (Circuit works CW2400). The measurement distance between any two adjacent antennas is 0.6 m, satisfied with far field requirement. The three antennas measurement method does not need a

calibrated reference antenna; the gain of the antennas can be solved after measuring $S_{21}$ with different combinations:

$$G_a = \frac{S_{21}^{ab} + S_{21}^{ac} - S_{21}^{bc} - 10\lg\left(\frac{4\pi d}{\lambda}\right)^2}{2}\,[\text{dB}], \qquad (4)$$

where $G_a$ is the measured graphene antenna gain, $S_{21}^{ab}$, $S_{21}^{ac}$, $S_{21}^{bc}$ are measured $S_{21}$ [in dB] between different antennas ($G_a$, $G_b$, $G_c$: printed graphene antenna). The last term describes free space loss and depends on wavelength $\lambda$ and distance $d$ between measured antennas.

Radiation patterns of the antenna were recorded using an antenna measurement system (Antenna Measurement Studio 5.5, Diamond Engineering) and N9918A VNA. The data were recorded every 2° rotation. Standard horn antenna was used as transmitting antenna. Different bands of horn antennas were used: C, X bands (Narda, standard gain horn, model 643, 642 and 640) and $K_u$ band (Steatite Q-par Antennas, standard horn, QSH18).

**RFID antenna measurement**. Read range and radiation pattern measurement was performed using a professional RFID measurement setup (Voyantic Tagformance Measurement System) with EIRP=4 W. Before measurement, the system was calibrated and the effects of receiver antenna gain, cable and free-space loss were eliminated. For read range measurement, the system sweeps linearly from 800 MHz to 1 GHz with 3 MHz steps. For radiation pattern measurement, the frequency is 915 MHz and data were recorded every 10° rotation.

**Antenna efficiency measurement**. The antenna total efficiency was measured by using rotary table, Antenna Measurement Studio 5.5 and N9918A VNA. The circuit conversion efficiency was measured using RF CW source (MARCONI, 6200) and precision multimeter (Agilent, U1251A).

## Data availability
The data that support the findings of this study are available from the corresponding author upon reasonable request.

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

## Acknowledgements

The project is supported by UK Engineering and Physical Research Council (EPN010345), the Royal Society, EU Graphene Flagship Program, European Research Council Synergy Grant Hetero2D, US Army Research Office (W911NF-16-1-0279) and NMS. We are grateful to Circa Group Pty Ltd and Dr. Rob McElroy (University of York) for providing Cyrene sample and technical advices.

## Author contributions

K.P. and Z.H. designed and prepared the experiments, measured and analysed the experimental data and drafted the manuscript. Y.F., T.L., J.L., J.Z. and Z.X. participated in measurements and drafted the manuscript. L.H. and J.G. drafted the manuscript. K.S.N. coordinated the project as well as drafted the manuscript. All authors have given approval to the final version of the manuscript.

## Additional information

**Competing interests:** The authors declare no competing interests.

