## [Peer Review File · Nature Communications]

Reviewers' comments:

Reviewer #1 (Remarks to the Author):

In this manuscript, the authors report the use of so-called graphene ink in cyrene, a bio-based solvent, for wireless connectivity applications. The main emphasis is on the use of graphene, and the cost which is claimed "to be low".

I have many concerns about this publication and I can only recommend to reject this manuscript for the following reasons:

1) The material described in this manuscript is definitely NOT graphene (as stated in the title and all along the manuscript) but multi-layer graphene (not even few-layer graphene), according to the classification proposed by Wick et al. (*Angewandte Chemie Int. Ed.*, 30, (2014), 7714). Graphene is a monoatomic thin single layer of carbon. With a mean thickness of 5nm, the number of layers is above 10.

Additional point: is there any temperature control during the 8-hour bath sonication step?

2) The cost of production of "graphene" ink using cyrene as a solvent is indeed a good idea, as already reported by the cited reference 29 (Salavagione et al. *Green Chem.* 19, (2017), 2550). However, it seems that the number of providers of this new solvent is rather low (I could find only one on the web, Sigma Aldrich) and the cost is rather high as a consequence (180€/L for 1L). Although cyrene is not significantly dangerous for health and rather benign towards the environment due to its high biodegradability, it is still rather irritant. With only one producer identified on the web (Circa group) and low volumes of production, cyrene is definitely NOT a low-cost solvent. Also, it is rather unlikely that 8-hour bath sonication can be used for large-scale production; also, centrifugation (even if the speed is not so high) is also not a classical industrial process. Finally, the complex series of centrifugations and filtrations which is describes doesn't look like a low-cost process.

3) There is no information in the document about the stability of the ink a concentration as high as 10mg/mL when cited reference 29 does not exceed 0.7mg/mL It is likely that at such a high concentration, the ink is more behaving like a rather fluid paste which cannot settle down because the concentration is too high, giving the false impression that it is stable The miscibility of cyrene with other organic solvents is unknown (this is not important here because cyrene is removed but this may be an issue in many other cases).

4) The protocols described for ink preparation and the characterisation of the electrical conductivity of the films are different (different centrifugation speeds), so it is likely that the material in suspension is different, and that the electrical conductivity is also different.

5) The IR spectroscopy data shown have not been processed correctly. The peaks at 2349 cm⁻¹ are badly corrected (appear as above or below the baseline depending on the spectrum); the presence of a large (-OH ?) band above 3500 cm⁻¹ and its absence in the spectrum of "graphene" suggests that the sample was too thick for transmission of the IR beam in the KBr pellet. So the absence of any significant peak for "graphene" cannot be used to claim that "the laminate is composed of largely defect-free material".

6) What is the meaning of "annealing" an ink at 100°C when the boiling point of cyrene is 227°C?

7) Regarding the part dealing with applications, I am afraid that the performances in terms of conductivity are very comparable or less than the state of the art (see for example *Materials Today* 21, (2018), 223).

Reviewer #2 (Remarks to the Author):

The manuscript proposes an original development of a cost-effective, high-conductivity graphene ink and explores its application to flexible antennas on paper for wireless connectivity of IoT devices. The results are very interesting for the scientific community since they demonstrate, through real-life applications, the effectiveness of the proposed approach. The manuscript is clear and scientifically sounded although there are minor issues in the method sections. For these motivations I suggest to accept the manuscript with minor reviews.

Specific points

1) Radiation pattern measurements.

With respect to equation (4):

* it should be specified that the scattering parameters S_{21} are expressed in dB;

* eqn. (4) is wrong in the present form: it is missed a factor 1/2 that multiplies all the second member terms. Please verify, correct and, if needed, update the gain graph (Figure 5b).

2) Energy harvesting antenna efficiency calculation.

* Equation (5) is wrong: numerator and denominator should be reversed: please verify and correct.

* It is not clear how conversion circuit efficiency is measured. I suggest do add details about the RF-to-DC conversion circuit and a measured value of its efficiency. Such an efficiency should, in turn, depends on the input RF power due to the diode non-linearities.

3) Figure 5i.

* It is not clear if the graph refers to the efficiency of the graphene antenna alone or to that of the whole RF energy harvesting circuit (antenna + diode rectifier). Please specify.

* Note that the antenna efficiency doesn't depends, in general, on the input power since antennas are linear, passive devices. In doing the computations with equation (5) the only residual dependence on the input power should be related to the matching between antenna and rectifier. This, of course, if the conversion circuit efficiency is correctly accounted for with its power dependent law.

Reviewer #3 (Remarks to the Author):

Dear authors

This paper gives a good description of synthesis of high concentration of graphene ink, using a non-toxic material synthesized from cellulose: Cyrene. As mentioned in the publication, this technique is well suitable for mass production, and gives could provide material with low resistivity (high conductivity). This kind of material are well adapted for screen printing technique.

The important results obtained is the ability to fabricate by screen-printing low cost antennas for applications in the radio frequency (RF) range such as energy harvesting, antennas for communication systems such as RFID, ... Therefore, these results are important for many readers

Here are some open questions and remarks:

Remark 1: Fig. (1-a) shows that sheet resistance could be at the same level considering 8H for cyrene treatment and 20 H for NMP solutions. One of the interesting comparison, from the reader point of view, is to see a comparison of antennas obtained with two approaches.

Remark 2: Fig. 1 d - e: In the figure label, the description of these results are inverted: "d" is related to the thickness while "e" is related to the "surface"

Remark 3: The deposition technique consist of deposition by screen printing, sintering, and compression. In this case, could author may give explanation about precision and resolution achieved on antennas for example ?

Reviewer #1 (Remarks to the Author):

In this manuscript, the authors report the use of so-called graphene ink in cyrene, a bio-based solvent, for wireless connectivity applications. The main emphasis is on the use of graphene, and the cost which is claimed "to be low".

I have many concerns about this publication and I can only recommend to reject this manuscript for the following reasons:

1) The material described in this manuscript is definitely NOT graphene (as stated in the title and all along the manuscript) but multi-layer graphene (not even few-layer graphene), according to the classification proposed by Wick et al. (Angewandte Chemie Int. Ed., 30, (2014), 7714). Graphene is a monoatomic thin single layer of carbon. With a mean thickness of 5nm, the number of layers is above 10.

Additional point: is there any temperature control during the 8-hour bath sonication step?

Ans: The major characteristics of any inks used for printable electronics is its conductivity, which needs to be made as high as possible. This is achieved by two parameters: making the printed inks as dense as possible (small fraction of voids) and by reducing the number of interconnects between flakes. The former one is achieved by the use of the thinnest possible flakes, and the latter – by using the flakes with the largest possible area. Any exfoliation method results in flakes with certain relation between their thickness and area: the thinner the flakes – the smaller the area. Thus, ideally, a distribution of flake sizes should be used, with a hierarchical structure of the printed ink where the large flakes provide the backbone and the smaller fill in the voids. The particular optimal distribution depends on many parameters, including the contact resistance and the conductivity of the individual flakes. In the manuscript we have provided detailed study of the flake thickness distribution we used. This distribution was chosen, because it was the most optimal for the contact resistance achieved with this particular solvent.

We completely agree that the distribution peaks at finite thickness of the flakes, which are not monolayer graphene. At the same time, we would like to stress, that the low-thickness tail of the distribution (where graphene nano-flakes are) is crucially important for the lowering of the overall resistance of the printed ink.

To make highly conductive graphene ink large contact areas between few- or multi-layer graphene flakes (flat surface) and high density are needed [1]. In fact, none of the liquid phase exfoliation methods can guarantee that all flakes dispersed in liquid is monolayer graphene [4][5][6], even in a very dilute dispersion (0.01mg/mL) [7] (apart of those which go through graphene oxide route, but those do not provide the conductivity required for the antenna applications). We carried out AFM measurement again (shown in Fig.1) and the mean flake thickness is 4.5 nm (868 flakes counted) which is very close to our previous results (about 5 nm). The graphene ink in [1] has mean flake thickness of 12 nm.

Fig. 1: (a) AFM images of Cyrene exfoliated graphene flakes on silicon substrate, (b) Graphene flakes identification (centre marked) and (c) Graphene flakes thickness distribution.

Obviously, low concentration graphene sheet dispersion is useless for high conductive printing [1], especially not suitable for screen printing because it requires appropriate viscosity. For higher concentration graphene ink, longer sonication time is required [3]. However, longer sonication time not only increases fabrication costs (consuming more energy) but also causes significant surface defects (lower G/D ratio in Raman spectra which degrades the electrical conductivity), which is not desirable in practical applications (such as consumer electronics) which require low-cost and high electrical conductivity. Other techniques, such as ‘microfluidization’ method can also achieve high concentration graphene ink for screen printing, but requiring 100 cycles of processes and having larger mean flake thickness (12 nm mean flake thickness) [1].

Furthermore, it is important to use the ink most suitable for the applications. Thin and small flakes allow for the best stacking, however, they end up with the maximum number of interfaces, which potentially can increase the resistance. Thick flakes allow one to reduce the number of interfaces between the flakes, but they do not guarantee good stacking and result in a number of voids when printed. We carefully chose the flake thickness to minimize the resistance (so to increase the electrical conductivity).

The water temperature during the operation was stabilized at 40°C (room temperature is around 25°C). There is a sensor inside the ultrasonic bath to monitor water temperature. If the temperature is higher than 50°C, the ultrasonic bath will be switched off. This is controlled by NI DAQ shown in Fig. 2 (b). The DAQ also controls the sonication time accordingly to pre-set value as the original bath could only run continuously for 1.5 hours.

Fig. 2: (a) Front view of the ultrasonic bath and (b) Top view of the ultrasonic bath.

2) The cost of production of "graphene" ink using cyrene as a solvent is indeed a good idea, as already reported by the cited reference 29 (Salavagione et al. Green Chem. 19, (2017), 2550). However, it seems that the number of providers of this new solvent is rather low (I could find only one on the web, Sigma Aldrich) and the cost is rather high as a consequence (180€/L for 1L). Although cyrene is not significantly dangerous for health and rather benign towards the environment due to its high biodegradability, it is still rather irritant. With only one producer identified on the web (Circa group) and low volumes of production, cyrene is definitely NOT a low-cost solvent. Also, it is rather unlikely that 8-hour bath sonication can be used for large-scale production; also, centrifugation (even if the speed is not so high) is also not a classical industrial process. Finally, the complex series of centrifugations and filtrations which is describes doesn't look like a low-cost process.

Ans:

1. Cyrene is a newly developed solvent and protected by patents. It's manufactured by Circa Group in Australia at moment. We are informed that they are establishing a production line for high volume Cyrene production. It's true that it's not cheap right now but it'll certainly become cheaper and cheaper as time goes. The fact that only one producer at moment doesn't mean that Cyrene must be expensive. Surely, Circa has its own business plan for maximally reducing their production costs so to provide Cyrene for potential lucrative conductive ink production worldwide.
2. Liquid phase exfoliation of graphene is a physical process. Cyrene can be reused after collection and simple filtration, further decreasing the costs.
3. Sonication treatment is a mature technology in industries. 8 hours sonication time is based on our lab capability. Higher sonication power could potentially reduce the sonication time (but this is out of the scope of this work). One of the main points in this work is to demonstrate that Cyrene solvent requires much less time than NMP or similar solvents, which significantly reduces the production costs.

4. Centrifugation and filtration are common and mature techniques used in industry (butter we eat is produced by centrifugation). It can be low-cost.
5. For printed flexible consumer electronics, such as NFC and UHF RFIDs, low production cost is of crucial importance. Current silver and copper nano-particle inks are far more expensive comparing to the technique proposed here (\$2000 per kg for silver ink [22] or \$1000 per kg for copper ink [23]).

3) There is no information in the document about the stability of the ink a concentration as high as 10mg/mL when cited reference 29 does not exceed 0.7mg/mL It is likely that at such a high concentration, the ink is more behaving like a rather fluid paste which cannot settle down because the concentration is too high, giving the false impression that it is stable The miscibility of cyrene with other organic solvents is unknown (this is not important here because cyrene is removed but this may be an issue in many other cases).

Ans: It is indeed that our ink is paste-like – exactly what is required for screen-printing [8]. Graphene ink looks like a normal liquid with the concentration of 10 mg/mL but like fluid paste when it's concentrated to 70 mg/mL. We've developed such ink because screen-printing is the method widely used for the structures we propose. We have included more information about the ink viscosity in supplementary information.

There is no surface charge on pure graphene sheets. Hence, aggregation and sedimentation phenomenons are more intense, especially in high concentration ink. The stability of highly concentrated graphene inks can be achieved by adding polymer assisted agents such as carboxymethyl cellulose (CMC) and CAB and surfactant assisted agent such as sodium deoxycholate (SDC). In [1] both CMC and SDC were used to stabilize the flakes against restacking for highly concentrated graphene ink. In our work CAB was added and absorbed on the surface of graphene flakes after sonication process, where electrostatic repulsion between CAB molecules can prevent graphene flakes from aggregation so to stabilize the ink [9]. We have made this clear in the revised version.

4) The protocols described for ink preparation and the characterisation of the electrical conductivity of the films are different (different centrifugation speeds), so it is likely that the material in suspension is different, and that the electrical conductivity is also different.

Ans: Many thanks for pointing this out. This must have been my mistake during merging the contributions from different participants. In fact, the protocols for ink preparation and the characterization of electrical conductivity of the film are exactly the same. They are:

Step 1. Filtering the samples via 300-mesh stainless steel screen.

Step 2. Centrifuging the samples for 5 mins with the speed of 500 rpm.

My apology for this mistake and thanks again for pointing this out.

5) The IR spectroscopy data shown have not been processed correctly. The peaks at 2349 cm⁻¹ are badly corrected (appear as above or below the baseline depending on the spectrum); the presence of a large (-OH ?) band above 3500 cm⁻¹ and its absence in the spectrum of "graphene" suggests that

the sample was too thick for transmission of the IR beam in the KBr pellet. So the absence of any significant peak for "graphene" cannot be used to claim that "the laminate is composed of largely defect-free material".

Ans: The peaks at 2300-2400 cm^{-1} is related to gaseous CO_2 in the air. Even the air background has been recorded and deducted from the samples, the peaks of CO_2 will still present at the spectra of the samples when the concentration of CO_2 changes during the measurement. When the concentration of CO_2 is higher than the recorded concentration, its peaks are below the baseline. Oppositely when the concentration of CO_2 decreased, its peaks are above the baseline. If the samples are not related to CO_2 , these peaks are not needed to be considered for analysis [10,11].

You are right that the large band above 3500 cm^{-1} in CAB and graphene-CAB in Fig. 2a is related to $-\text{OH}$, being the intramolecular hydrogen bonds of CAB [12]. It doesn't belong to graphene.

For graphene oxide (GO), even reduced graphene oxide (rGO), there are some functional groups on the graphene sheets, such as $-\text{OH}$, $-\text{COOH}$, and so on. Therefore, there are some peaks for the FTIR spectra of GO and rGO. But for graphene by exfoliation of graphite, there are only C atoms, no $-\text{OH}$ and $-\text{COOH}$ groups [13]. Therefore, there are no peaks at 3500 cm^{-1} (3490 cm^{-1} precisely), 1340 cm^{-1} for $-\text{OH}$ and 1710-1720 cm^{-1} for $-\text{COOH}$, respectively [14]. This was also verified by graphene nanoribbons which were formed by longitudinal unzipping of carbon nanotubes [15], graphene produced by electrolytic exfoliation [16] and organic salt-assisted liquid-phase exfoliation [17] from graphite. For pristine graphite, its FTIR spectrum is featureless, as reported in [18-20]. Therefore pristine graphene naturally has only a very weak peak for $\text{C}=\text{C}$ at $\sim 1580 \text{ cm}^{-1}$ [17]. Except $\text{C}=\text{C}$, graphene has no other IR absorptions no matter what the thickness of the IR sample is. In other word, the absence of peaks of graphene is not related to the thickness of the KBr pellet.

Pristine graphene has no other IR peaks except $\text{C}=\text{C}$. Only if graphene is oxidized or functionalized, there are some corresponding peaks on IR spectrum for various chemical groups, such as $-\text{OH}$, $-\text{COOH}$, and so on. These chemical groups are regarded as defects. Conversely, when no peaks are detected by IR for a graphene sample, it is reasonable to claim this sample is composed of largely defect-free material.

6) What is the meaning of "annealing" an ink at 100°C when the boiling point of cyrene is 227 °C?

Ans: Cyrene will decompose when the temperature is higher than 200°C and should be avoided to use it above 140 °C [21]. Therefore, we annealed the printed pattern at 100 °C in vacuum (the boiling point of Cyrene is 116°C/10mBar [21]). At 100 °C Cyrene evaporates, reducing the resistance between the flakes. We are trying to avoid harsh evaporation; as such process produces bubbles and voids, increasing the overall resistance.

7) Regarding the part dealing with applications, I am afraid that the performances in terms of conductivity are very comparable or less than the state of the art (see for example *Materials Today* 21, (2018), 223).

Ans: The work reported in *Materials Today* is not related to graphene ink. The technique reported can't be used to print electronic components such as antennas. The work reported did not produce

any graphene ink. The graphene paper was realized by compressing the graphene nano-platelet powder at 200 Bar at room temperature. High compression pressure can recrystallize carbon atoms so to increase the film conductivity. The antenna patterns were cut by laser ablation or blade cutting (NOT printed!), which significantly limits its applications for printed flexible RF electronics. The technique reported in our work is completely different: we report a high conductive graphene ink and antennas for wireless connectivity and IoT applications.

Graphene ink has many advantages in comparison with other printable inks. It is flexible, inexpensive, environmentally friendly, etc.

References:

- [1] Karagiannidis, Panagiotis G., et al. "Microfluidization of graphite and formulation of graphene-based conductive inks." *ACS Nano* 11.3 (2017): 2742-2755.
- [2] Ciesielski, Artur, and Paolo Samori. "Graphene via sonication assisted liquid-phase exfoliation." *Chemical Society Reviews* 43.1 (2014): 381-398.
- [3] Khan, Umar, et al. "High concentration solvent exfoliation of graphene." *Small* 6.7 (2010): 864-871.
- [4] Secor, Ethan B., et al. "Rapid and versatile photonic annealing of graphene inks for flexible printed electronics." *Advanced Materials* 27.42 (2015): 6683-6688.
- [5] Secor, Ethan B., et al. "Enhanced Conductivity, Adhesion, and Environmental Stability of Printed Graphene Inks with Nitrocellulose." *Chemistry of Materials* 29.5 (2017): 2332-2340.
- [6] Dong, Lei, et al. "A non-dispersion strategy for large-scale production of ultra-high concentration graphene slurries in water." *Nature communications* 9.1 (2018): 76.
- [7] Hernandez, Yenny, et al. "High-yield production of graphene by liquid-phase exfoliation of graphite." *Nature nanotechnology* 3.9 (2008): 563.
- [8] Khan, Saleem, Leandro Lorenzelli, and Ravinder Dahiya. "Technologies for printing sensors and electronics over large flexible substrates: a review." *IEEE Sensors Journal* 15.6 (2014): 3164-3185.
- [9] Ciesielski, Artur, and Paolo Samori. "Graphene via sonication assisted liquid-phase exfoliation". *Chemical Society Reviews* 43,381-398 (2014).
- [10] Rege, Salil U., and Ralph T. Yang. "A novel FTIR method for studying mixed gas adsorption at low concentrations: H₂O and CO₂ on NaX zeolite and γ -alumina." *Chemical Engineering Science* 56.12 (2001): 3781-3796.
- [11] Stevens Jr, Robert W., Ranjani V. Siriwardane, and Jennifer Logan. "In situ Fourier transform infrared (FTIR) investigation of CO₂ adsorption onto zeolite materials." *Energy & Fuels* 22.5 (2008): 3070-3079.
- [12] Oh, Sang Youn, et al. "Crystalline structure analysis of cellulose treated with sodium hydroxide and carbon dioxide by means of X-ray diffraction and FTIR spectroscopy." *Carbohydrate research* 340.15 (2005): 2376-2391.
- [13] Lotya, Mustafa, et al. "Liquid phase production of graphene by exfoliation of graphite in surfactant/water solutions." *Journal of the American Chemical Society* 131.10 (2009): 3611-3620.
- [14] Si, Yongchao, and Edward T. Samulski. "Synthesis of water soluble graphene." *Nano letters* 8.6 (2008): 1679-1682.
- [15] Kosynkin, Dmitry V., et al. "Longitudinal unzipping of carbon nanotubes to form graphene nanoribbons." *Nature* 458.7240 (2009): 872.

- [16] Wang, Guoxiu, et al. "Highly efficient and large-scale synthesis of graphene by electrolytic exfoliation." *Carbon* 47.14 (2009): 3242-3246.
- [17] Du, Wencheng, et al. "Organic salt-assisted liquid-phase exfoliation of graphite to produce high-quality graphene." *Chemical Physics Letters* 568 (2013): 198-201.
- [18] Guo, Hui-Lin, et al. "A green approach to the synthesis of graphene nanosheets." *ACS Nano* 3.9 (2009): 2653-2659.
- [19] Liu, Yi-Tao, Xu-Ming Xie, and Xiong-Ying Ye. "High-concentration organic solutions of poly (styrene-co-butadiene-co-styrene)-modified graphene sheets exfoliated from graphite." *Carbon* 49.11 (2011): 3529-3537.
- [20] Liu, Na, et al. "One - step ionic - liquid - assisted electrochemical synthesis of ionic - liquid - functionalized graphene sheets directly from graphite." *Advanced Functional Materials* 18.10 (2008): 1518-1525.
- [21]https://static1.squarespace.com/static/5643eb3de4b0c236a2510a8c/t/5922423c5016e1a64b943dda/1495417406168/Datasheet_Cyrene170306%5B7%5D.pdf
- [22]<http://store.novacentrix.com/HPS-021LV-silver-screen-print-ink-p/910-0081-01.htm>
- [23]<http://store.novacentrix.com/Metalon-ICI-021-Copper-oxide-screen-print-ink-p/910-0095-01.htm>

Reviewer #2 (Remarks to the Author):

The manuscript proposes an original development of a cost-effective, high-conductivity graphene ink and explores its application to flexible antennas on paper for wireless connectivity of IoT devices. The results are very interesting for the scientific community since they demonstrate, through real-life applications, the effectiveness of the proposed approach. The manuscript is clear and scientifically sounded although there are minor issues in the method sections. For these motivations I suggest to accept the manuscript with minor reviews.

Specific points

1) Radiation pattern measurements.

With respect to equation (4):

* it should be specified that the scattering parameters S_{21} are expressed in dB;

* eqn. (4) is wrong in the present form: it is missed a factor 1/2 that multiplies all the second member terms. Please verify, correct and, if needed, update the gain graph (Figure 5b).

Ans: Thanks for pointing this out. We have specified the S_{21} in dB and corrected the equation accordingly. The gain graph (Figure 5b in the main text) is correct because the gain was directly measured using the 3-antennas measurement method (Antenna Measurement Studio 5.5 from Diamond Engineering and Agilent N9918A VNA) [1].

2) Energy harvesting antenna efficiency calculation.

* Equation (5) is wrong: numerator and denominator should be reversed: please verify and correct.

* It is not clear how conversion circuit efficiency is measured. I suggest do add details about the RF-to-DC conversion circuit and a measured value of its efficiency. Such an efficiency should, in turn, depends on the input RF power due to the diode non-linearities.

Ans: Thanks for pointing it out. We have corrected accordingly.

Fig. 1: Energy flow diagram of the energy harvesting system.

Referring to Fig.1, the overall efficiency can be expressed as $\eta_{overall} = \frac{P_{DC}}{P_{Ant-in}} = \eta_{ant-total} \times$

$\eta_{circuit}$. It depends on (a) antenna total efficiency $\eta_{ant-total} = \frac{P_{Circuit-in}}{P_{Ant-in}}$ and (b) AC-to-DC

conversion efficiency $\eta_{circuit} = \frac{P_{DC}}{P_{Circuit-in}}$. The purpose of the low pass filter connecting antenna

and rectifier is to prevent harmonic radiation. In this work, the antenna was matched to 50 Ohm.

The conversion circuit (4-stage Cockcroft–Walton voltage multiplier) was also matched to 50 Ohm at small signal level (-10 dBm). The overall efficiency does vary with RF input power due to diode

non-linearity. We have added details about the measured overall efficiency and measured AC-to-DC conversion efficiency in the paper.

3) Figure 5i.

* It is not clear if the graph refers to the efficiency of the graphene antenna alone or to that of the whole RF energy harvesting circuit (antenna + diode rectifier). Please specify.

* Note that the antenna efficiency doesn't depend, in general, on the input power since antennas are linear, passive devices. In doing the computations with equation (5) the only residual dependence on the input power should be related to the matching between antenna and rectifier. This, of course, if the conversion circuit efficiency is correctly accounted for with its power dependent law.

Ans: We have made this clear in the revised version by adding overall efficiency, AC-to-DC rectifier conversion efficiency and antenna total efficiency in Fig. 5i. There was also an error in the original graph which has been corrected in the revised version.

References:

[1] 'Antenna gain measurement using the 3-point method- Does not require a calibrated reference antenna' Diamond Engineering. <http://www.diamondeng.net>

Reviewer #3 (Remarks to the Author):

Dear authors

This paper gives a good description of synthesis of high concentration of graphene ink, using a non-toxic material synthesized from cellulose: Cyrene. As mentioned in the publication, this technique is well suitable for mass production, and gives could provide material with low resistivity (high conductivity). This kind of material are well adapted for screen printing technique.

The important results obtained is the ability to fabricate by screen-printing low cost antennas for applications in the radio frequency (RF) range such as energy harvesting, antennas for communication systems such as RFID, ... Therefore, these results are important for many readers

Here are some open questions and remarks:

Remark 1: Fig. (1-a) shows that sheet resistance could be at the same level considering 8Hfor cyrene treatment and 20 H for NMP solutions. One of the interesting comparison, from the reader point of view, is to see a comparison of antennas obtained with two approaches.

Ans: We are grateful for this very good advice and very much appreciated. We've printed the same ultra-wideband slot antenna but using NMP based graphene ink (20h ultrasonic treatment). The comparison of the measured results of these two antennas has been added in supplementary materials and shown below. It can be seen that the 8h Cyrene based antenna has slightly higher gain than 20h NMP antenna because the conductivity of 8h Cyrene printed pattern is slightly higher.

Fig. 1: Antenna gain based on different graphene inks.

Remark 2: Fig. 1 d - e: In the figure label, the description of these results are inverted: “d” is related

to the thickness while “e” is related to the “surface”

Ans: Thanks for pointing them out. We have corrected them accordingly.

Remark 3: The deposition technique consist of deposition by screen printing, sintering, and compression. In this case, could author may give explanation about precision and resolution achieved on antennas for example?

Ans: The resolution we can do now is about 0.4 mm. We have added this in the revised manuscript. There is little change of dimensions during sintering and compression process (approx. few micrometers).

REVIEWERS' COMMENTS:

Reviewer #1 (Remarks to the Author):

After reading the answers given to my questions by the authors I see that great care was given to justify them. I am happy to say that I am satisfied with most of them, except 2: although I understand the given detailed explanations, I still think that the title should be modified because at the time of writing this manuscript, the ink is NOT containing "graphene" but "multilayer graphene" in the best case, and the cost is NOT low at the moment (even if it may become low "in the future", this manuscript is not dealing with this issue and I thus see no reason to have this emphasized in the title).

I would thus recommend to modify the title accordingly before the manuscript can be accepted for publication.

Reviewer #2 (Remarks to the Author):

The authors have improved their manuscript by addressing all the reviewer's comments and suggestions. In my opinion the paper can be published as it is

Reviewer #3 (Remarks to the Author):

Dear Authors

The second version of this manuscript is quite clear. All the constructive remarks of the reviewers were taken into account to improve this interesting work based on new approach for synthesis of graphene ink and its applications.

Based on this, I suggest to accept the manuscript for publication.

Responses to Reviewers' Comments:

Reviewer #1 (Remarks to the Author):

After reading the answers given to my questions by the authors I see that great care was given to justify them. I am happy to say that I am satisfied with most of them, except 2: although I understand the given detailed explanations, I still think that the title should be modified because at the time of writing this manuscript, the ink is NOT containing "graphene" but "multilayer graphene" in the best case, and the cost is NOT low at the moment (even if it may become low "in the future", this manuscript is not dealing with this issue and I thus see no reason to have this emphasized in the title). I would thus recommend to modify the title accordingly before the manuscript can be accepted for publication.

Ans: We have changed the title according to your suggestion. The new title is 'Sustainable Production of Highly Conductive Multilayer Graphene Ink for Wireless Connectivity and IoT Applications'.

Reviewer #2 (Remarks to the Author):

The authors have improved their manuscript by addressing all the reviewer's comments and suggestions. In my opinion the paper can be published as it is

Many thanks.

Reviewer #3 (Remarks to the Author):

Dear Authors

The second version of this manuscript is quite clear. All the constructive remarks of the reviewers were taken into account to improve this interesting work based on new approach for synthesis of graphene ink and its applications.

Based on this, I suggest to accept the manuscript for publication.

Many Thanks.